# MonoCon: A general framework for learning ultra-compact high-fidelity representations using monotonicity constraints

## Abstract

Learning high-quality, robust, efficient, and disentangled representations is a central challenge in artificial intelligence (AI). Deep metric learning frameworks tackle this challenge primarily using architectural and optimization constraints. Here, we introduce a third approach that instead relies on *functional* constraints. Specifically, we present MonoCon, a simple framework that uses a small monotonic multi-layer perceptron (MLP) head attached to any pre-trained encoder. Due to co-adaptation between encoder and head guided by contrastive loss and monotonicity constraints, MonoCon learns robust, disentangled, and highly compact embeddings at a practically negligible performance cost. On the CIFAR-100 image classification task, MonoCon yields representations that are nearly 9x more compact and 1.5x more robust than the fine-tuned encoder baseline, while retaining 99% of the baseline's 5-NN classification accuracy. We also report a 3.4x more compact and 1.4x more robust representation on an SNLI sentence similarity task for a marginal reduction in the STSb score, establishing MonoCon as a general domain-agnostic framework. Crucially, these robust, ultra-compact representations learned via functional constraints offer a unified solution to critical challenges in disparate contexts ranging from edge computing to cloud-scale retrieval.

## 1 Introduction

Representation learning aims to extract meaningful features from raw data to enable downstream tasks such as prediction and classification. Deep metric learning techniques achieve such automated feature extraction by learning geometric relationships between data points based on semantic similarity measures Kaya & Bilge (2019). These structured representations, or embeddings, can be used for a variety of applications ranging from face recognition Schroff et al. (2015), image retrieval Babenko & Lempitsky (2015), and recommendation systems Covington et al. (2016). While undoubtedly powerful, the high dimensionality of learned embeddings entails significant costs in terms of storage, latency, and computational power Wang et al. (2021).

Efforts to mitigate these costs can be broadly divided into approaches focusing on representation compactness and model compactness. Representation compactness can be achieved using simple post-hoc dimensionality reduction using principal component analysis (PCA), but this typically incurs a significant performance cost Babenko & Lempitsky (2015). Fixed architectural bottlenecks, such as in autoencoders Hinton & Salakhutdinov (2006), provide explicit control over representation dimensionality but can be too restrictive or too wasteful depending on the context, and may require extensive hyperparameter tuning. Optimization-based solutions such as activation sparsity regularization Glorot et al. (2011) and Vicreg loss Bardes et al. (2021) provide more implicit control by engineering the loss function to promote feature sparsity and disentanglement. Representation quantization, a form of lossy compression, reduces the computational footprint of downstream tasks by reducing the number of bits used to store embeddings Jegou et al. (2010). Contemporary approaches to model compactness include L1 regularization Han et al. (2015), which prunes neural networks by setting unnecessary weights to zero, and knowledge distillation Hinton et al. (2015), where a smaller "student" network mimics a larger "teacher", often at a small performance cost.

Finally, quantization has also been implemented to reduce model size and improve inference speed Jacob et al. (2018).

Methods to achieve representation compactness almost exclusively utilize architectural or optimization constraints. Here, we introduce MonoCon, a new framework that instead leverages the *functional* constraint of monotonicity to learn robust and ultra-compact representations while retaining high performance. Operationally, MonoCon is based on a simple modification of standard metric learning frameworks: attaching a small monotonic MLP head to a pre-trained feature encoder, and training the model end-to-end to minimize supervised contrastive (SupCon) loss. We hypothesized that the mathematical incompatibility between monotonicity constraints and strong anti-correlations between the encoders' features could force the monotonic MLP to prune conflicting features, leading to more compact representations. Our experiments confirm this hypothesis and reveal a richer suite of findings that are summarized below:

- MonoCon achieves $\sim 8.9$x reduction in effective dimensionality, defined as the number of PCA components required to explain 99% variance of the training dataset, on the CIFAR-100 image classification task. It also achieves 1.5x error reduction in PCA-based reconstruction of test embeddings, while retaining $\sim 99\%$ baseline 5-NN classification accuracy.

- MonoCon intelligently adapts to data complexity, evidenced by a 7 dimensional native representation for CIFAR-10 compared to a 14 dimensional one for CIFAR-100.

- We demonstrate MonoCon's domain-agnostic nature on the SNLI sentence similarity task, where it learns 3.4x more compact and 1.4x more robust representations relative to the baseline at a practically negligible performance cost.

- Analysis of MonoCon's output feature correlation matrices reveals emergent block diagonal structure, demonstrating that the model's efficiency and robustness stem from a disentangled and modular representation.

- Training dynamics reveal complex self-organized co-adaptation between encoder and head characterized by a rapid initial dimensional collapse and gradual recovery, a process we term "embedding distillation".

## 2 RELATED WORK

### 2.1 DEEP METRIC LEARNING

The fundamental goal of deep metric learning is to train an encoder network to learn an embedding function that maps raw data points to a semantically structured space using a simple organizing principle: pull similar data points towards each other and push dissimilar ones further apart. From the original contrastive loss that used pairs of positive (similar) and negative (dissimilar) data points Hadsell et al. (2006), researchers have devised a diverse array of loss functions including Triplet Schroff et al. (2015), Barlow twins Zbontar et al. (2021), Vicreg Bardes et al. (2021), Multi-similarity Wang et al. (2019), InfoNCE Oord et al. (2018), and SupCon Khosla et al. (2020), to better structure the learned embeddings. While positive and negative pairs are easiest to define in supervised frameworks using labeled datasets, deep metric learning can also be implemented in a self-supervised setting Mikolov et al. (2013); He et al. (2020); Chen et al. (2020). Finally, the idea of using a small disposable MLP projection head has been shown to be powerful in further improving the quality of learned embeddings Chen et al. (2020).

### 2.2 MODEL AND REPRESENTATION COMPACTNESS

An important milestone in the quest for compact models is knowledge distillation Hinton et al. (2015), a process whereby a smaller "student" network learns to mimic the performance of a larger "teacher" network. Following earlier pioneering work on computer vision tasks Romero et al. (2014); Hinton et al. (2015); Yim et al. (2017), knowledge distillation has been used extensively in the natural language (NL) domain to create distilled versions of foundation models, such as distil-BERT Sanh et al. (2019) and TinyBERT Jiao et al. (2019), as well as in sentence encoders Reimers & Gurevych (2020). Other useful strategies to ensure compactness include pruning redundant weights and neurons Han et al. (2015); Frankle & Carbin (2018), low-rank factorization Denil et al. (2013),

model quantization Jacob et al. (2018), and intrinsically efficient architectural designs Howard et al. (2017); Iandola et al. (2016).

The most straightforward way to achieve representation compactness is by imposing a rigid architectural bottleneck on latent space dimension, as is done in autoencoders Hinton & Salakhutdinov (2006) and their variants, such as denoising Vincent et al. (2008) and variational autoencoders Kingma & Welling (2013). A flexible and highly popular approach is to include regularization terms in the loss function that encourage the model to learn representations with desirable traits such as compactness and feature disentanglement. These specialized loss functions include activation sparsity regularization Glorot et al. (2011), VICreg Bardes et al. (2021), Barlow twins Zbontar et al. (2021), InfoNCE Oord et al. (2018), mutual information maximization Hjelm et al. (2018), and variational information bottleneck (VIB) Alemi et al. (2016). Moreover, learned representations can be further compressed using post-hoc processing steps such as PCA-based truncation Babenko & Lempitsky (2015) and product quantization Jegou et al. (2010).

### 2.3 MONOTONIC NEURAL NETWORKS

Historically, monotonic neural networks were designed as a means of incorporating prior domain-specific knowledge of monotonic relationships between variables into machine learning tasks Sill (1997). In modern deep learning, monotonic neural networks play a crucial role in ensuring fairness and interpretability while applying machine learning techniques to high-stakes applications You et al. (2017); Liu et al. (2020); Gupta et al. (2016). The simplest way to build monotonic MLPs is to use non-negative weights and non-decreasing activation functions in the model's forward pass Wehenkel & Louppe (2019). However, monotonic neural networks is an active area of research, and new architectures continue to be proposed and implemented Runje & Shankaranarayana (2023); Kitouni et al. (2023); Kim & Lee (2024).

In summary, deep metric learning and monotonic neural networks constitute distinct and mature fields motivated by disparate goals: high-quality representations for the former, and fairness and interpretability for the latter. However, the potential of utilizing the functional constraint of monotonicity to induce representational compactness and disentanglement via emergent self-organization in deep metric learning models has, to the best of our knowledge, remained unexplored. The present work therefore represents a unique synthesis of two pervasive deep learning methodologies.

## 3 METHODOLOGY

### 3.1 SUPERVISED CONTRASTIVE LEARNING

We train MonoCon within a supervised contrastive learning setting. Contrastive learning shapes representations by pulling similar samples together and dissimilar ones far apart. In a supervised setting, this is implemented by treating samples with the same class label as positive pairs, and those with different class labels as negative pairs. We minimize the supervised contrastive loss function Khosla et al. (2020)

$$\mathcal{L}_{\text{SupCon}} = \sum_{i \in I} \frac{-1}{|P(i)|} \sum_{p \in P(i)} \log \frac{\exp\left(\mathbf{z}_i \cdot \mathbf{z}_p / \tau\right)}{\sum_{a \in A(i)} \exp\left(\mathbf{z}_i \cdot \mathbf{z}_a / \tau\right)} \tag{1}$$

where $I$ is the set of all indices in a mini-batch, $i$ is the index of the anchor sample, $\mathbf{z}_i$ is the normalized output embedding vector of the model for anchor $i$, $A(i)$ is the set of all indices in the batch excluding the anchor $i$, $P(i)$ is the set of indices for all positives for anchor $i$ in the mini-batch, $|P(i)|$ is the total number of positives, and $\tau$ is the "temperature" hyperparameter that controls class separation.

### 3.2 MONOTONIC MLP HEAD

The central innovation of MonoCon is the monotonic MLP head. This module provides a strong inductive bias via its functional monotonicity constraints, which is critical for producing the well-structured, compact, and robust embeddings achieved in this work. For all experiments in this work,

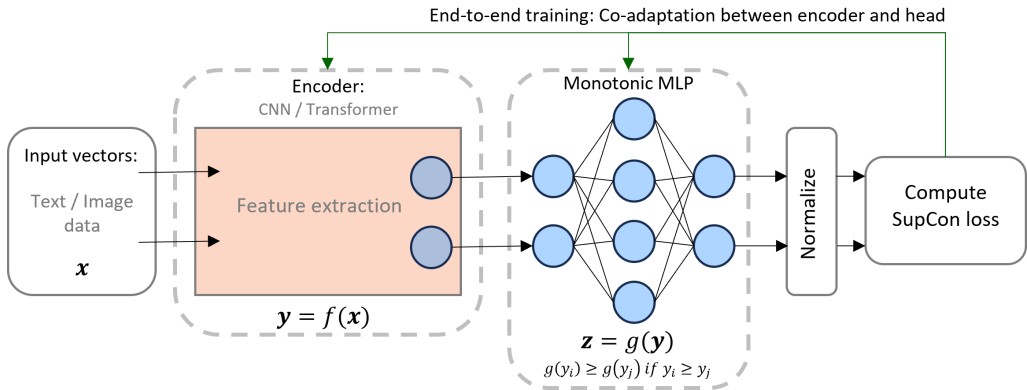

Figure 1: **Block diagram of MonoCon:** Co-adaptation between a feature encoder and a non-disposable monotonic MLP head in a supervised contrastive learning setup.

the monotonic MLP head is chosen to have a single hidden layer. The input and output dimensions of the monotonic MLP match the output embedding dimension of the encoder $d_{enc}$, and the width of the hidden layer is $2d_{enc}$. We implement monotonicity constraints by squaring the weights to ensure non-negativity, and using the non-decreasing Leaky ReLU activation function.

### 3.3 MONOCON ARCHITECTURE

MonoCon's architecture is remarkably simple, as shown in the block diagram in Fig. 1. The monotonic MLP head is attached to a pre-trained encoder, and the model is trained end-to-end using SupCon loss (Eq. 1). We implement a differential learning rate strategy to flexibly switch between fine-tuning the encoder at a lower rate, to full-fledged co-adaptation between encoder and head at the same rate. MonoCon's philosophy requires the encoder to have sufficient flexibility to simultaneously minimize contrastive loss and respect monotonicity constraints, for which fine-tuning at a small learning rate may not be sufficient. It is worth noting a couple of important differences between the monotonic MLP head of MonoCon and the projection head of SimCLR Chen et al. (2020). In SimCLR, the projection head is a standard MLP with a smaller output dimension than the encoder, whereas in MonoCon, the output dimension of the head is the *same* as that of the encoder. Most crucially, in SimCLR, the projection head is discarded at the end of training, and its role is to improve the quality of the encoder's embeddings. In stark contrast, output vectors of the monotonic MLP head *are* the final efficient and high-fidelity embeddings of MonoCon.

## 4 EXPERIMENTAL SETUP

### 4.1 DATASETS

For vision tasks, we imported the CIFAR-10 and CIFAR-100 benchmark datasets Krizhevsky et al. (2009) from torchvision. CIFAR-10 has 10 labeled classes whereas CIFAR-100 has 100. Both datasets contain 50,000 images at 32x32 pixel resolution. Our dataset creation and data loading pipelines are identical across CIFAR-10 and CIFAR-100 experiments. We trained the model using augmented versions of the CIFAR training datasets using the TrivialAugmentWide Müller & Hutter (2021) and RandomErasing Zhong et al. (2020) transforms from torchvision. For validation, we created gallery and query subsets using 90% and 10% of the unaugmented training dataset, respectively. For the NL task, we used the Stanford Natural Language Inference (SNLI) database Bowman et al. (2015) to train our model and the Semantic Textual Similarity Benchmark (STSb) Cer et al. (2017) for validation, using the Spearman correlation coefficient as our metric.

### 4.2 BASELINES

To demonstrate that MonoCon can learn compact, robust, and disentangled representations while providing high performance, we performed ablation studies to compare MonoCon with powerful

fine-tuned encoder baselines without the monotonic MLP head. For vision tasks, we fine-tuned a pre-trained ResNet34 encoder He et al. (2016) using SupCon loss. The first convolutional and max pooling layers were suitably adapted for low-resolution CIFAR images, and the final classification layer was removed. For the NL task, we fine-tuned a pre-trained all-MiniLM-L6-v2 sentence encoder Wang et al. (2020) using SupCon loss, using sentence pairs with an entailment relationship as positive pairs.

### 4.3 EVALUATION METRICS

For vision tasks, we used 5-NN classification accuracy and Recall@1 as our validation metrics and computed them every 5 epochs during training. For the NL task, we computed the Spearman correlation coefficient at the end of every epoch for validation. To quantify representation compactness, we defined their effective dimensionality ($d_{\text{eff}}$) as the number of PCA components required to explain at least 99% variance of the training dataset. We define representation robustness as the fidelity with which a model's representation can reconstruct embeddings for unseen data, and quantify it using the root mean squared (RMS) reconstruction error for test embeddings projected onto the PCA space of the train embeddings. To quantify and visualize the structure of learned representations, we computed feature correlation matrices based on output embeddings of the encoder as well as monotonic MLP head, and generated the corresponding clustermaps and dendrograms using standard agglomerative clustering algorithms.

Please refer to Appendix A for implementation details, Appendix B for hyperparameter choices, and Appendix C for an LLM usage statement.

## 5 RESULTS

### 5.1 MONOCON PRODUCES ULTRA-COMPACT HIGH-FIDELITY REPRESENTATIONS ACROSS VISION AND LANGUAGE TASKS

The most compelling feature of MonoCon is its ability to produce remarkably low dimensional representations at a marginal performance cost. As shown in Table 1, MonoCon performs comparably to the fine-tuned ResNet34 baseline on CIFAR-100 image classification, suffering only a marginal 0.74% drop in 5-NN accuracy (See Fig. 5 for tSNE plots) and 2.58% in Recall@1, and even slightly outperforming the baseline on Recall@5. Remarkably, it achieves this performance while providing 88.8% reduction in dimensionality and 35.5% reduction in PCA reconstruction error. This simultaneous increase in compactness and robustness is observed on CIFAR-10 image classification as well as the SNLI sentence similarity task, suggesting that MonoCon inherently learns semantically well-organized representations. This claim is further supported by the variation of Recall@k with $k$, which shows a clear crossover from the baseline winning for $k < 3$ to MonoCon winning for $k > 3$ (Fig. 6). This suggests that while the baseline specializes in precision retrieval, MonoCon has a better global semantic structure and hence better semantic neighborhoods.

### 5.2 MONOTONICITY CONSTRAINTS ARE ESSENTIAL FOR PRODUCING ROBUST AND EFFICIENT REPRESENTATIONS

While the differences between MonoCon and the baseline are clearly due to the MLP head, it is important to ascertain whether the efficiency and robustness follow from the presence of an MLP head, or specifically from the constraint of monotonicity. To answer this question, we performed an ablation study for the CIFAR-100 task, by replacing the monotonic MLP head with a standard MLP with an identical architecture and Leaky ReLU activation, but without enforcing non-negativity on neural weights. As shown in Table 2, using a standard MLP head leads to improvement over the baseline on all metrics except Recall@1. However, $d_{\text{eff}} = 78$, while considerably lower than the baseline's $d_{\text{eff}} = 125$, is still 5.5x larger than MonoCon's $d_{\text{eff}} = 14$. Furthermore, while the PCA reconstruction error for MonoCon and standard MLP head are comparable, MonoCon requires 5.5x fewer dimensions to achieve the same level of robustness. Thus, our ablation study conclusively demonstrates that MonoCon's massive gains in robustness and efficiency are a direct result of its monotonicity constraints.

Table 1: Comparison of MonoCon against a fine-tuned baseline on vision and natural language tasks.

(a) Performance summary for vision tasks.

| Dataset | Model | 5-NN Acc (% ↑) | Recall@1 (% ↑) | Recall@5 (% ↑) | Effective dim. $d_{eff}$ ↓ | PCA recon. error with $d_{eff}$ dims. ($\times 10^{-3}$ ↓) |
|---|---|---|---|---|---|---|
| CIFAR-100 | Baseline | 77.75 | 76.63 | 82.71 | 125 | 6.40 |
| | MonoCon | 77.01 | 74.05 | 83.31 | **14** | **4.13** |
| CIFAR-10 | Baseline | 94.36 | 93.80 | 96.84 | 21 | 3.75 |
| | MonoCon | 94.54 | 93.19 | 97.24 | **7** | **3.22** |

(b) Performance summary for NL sentence similarity task.

| Dataset | Model | STSb score (% ↑) | Effective dim. $d_{eff}$ ↓ | PCA recon. error with $d_{eff}$ dims. ($\times 10^{-3}$ ↓) |
|---|---|---|---|---|
| SNLI | Baseline | 81.78 | 292 | 9.74 |
| | MonoCon | 81.25 | **86** | **6.92** |

Table 2: Performance of standard MLP head compared to Baseline and MonoCon for CIFAR-100.

| Model | 5-NN Acc (% ↑) | Recall@1 (% ↑) | Recall@5 (% ↑) | Effective dim. $d_{eff}$ ↓ | PCA recon. error with $d_{eff}$ dims. ($\times 10^{-3}$ ↓) |
|---|---|---|---|---|---|
| Baseline | 77.75 | 76.63 | 82.71 | 125 | 6.40 |
| MonoCon | 77.01 | 74.05 | 83.31 | **14** | **4.13** |
| Std. MLP head | 78.32 | 76.04 | 83.48 | 78 | 4.15 |

## 5.3 MONOCON'S COMPACTNESS AND ROBUSTNESS FOLLOW FROM ITS DISENTANGLED REPRESENTATION

The combination of strong performance, ultra-compactness, and low reconstruction error suggests that MonoCon's representation contains highly organized and disentangled features. To investigate this possibility, we computed Pearson correlation matrices for normalized output embeddings of the models analyzed in Table 2, using a fixed subset of 5000 images from the training dataset (Fig. 2). Clustermaps of these matrices show that the baseline and standard MLP head models learn entangled representations, as evidenced by weak and diffuse correlations and unstructured dendrograms. By stark contrast, MonoCon's feature correlation matrix shows highly pronounced block diagonal structure, which is mirrored by a clear hierarchy in the dendrogram. We observed a similar pattern in MonoCon's representation for CIFAR-10 and SNLI tasks as well (Fig. 7). This structure demonstrates a systematic division of features into distinct groups with strong intra-group correlations and weak inter-group correlations. Thus, MonoCon achieves a sophisticated form of disentanglement at the level of higher order "concepts" associated with correlated feature groups, rather than individual features Zbontar et al. (2021); Bardes et al. (2021).

## 5.4 MONOCON DELIVERS HIGH PERFORMANCE UNDER EXTREME REPRESENTATION COMPRESSION

MonoCon's compact and disentangled representation makes it a potentially superior choice under high levels of representation compression. High-quality compressed representations are vital not only in resource-constrained environments like edge devices, but also for reducing costs and improving the speed of large-scale search and retrieval. We therefore compared the performance of MonoCon, baseline, and standard MLP head models on CIFAR-100 by truncating their represen-

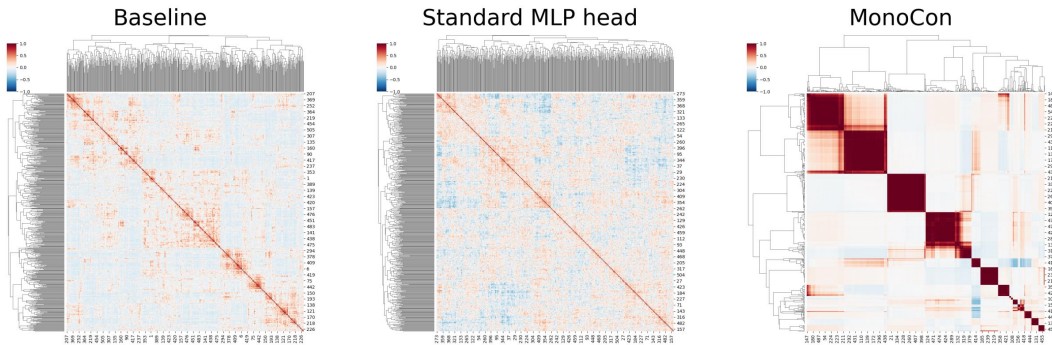

Figure 2: **MonoCon learns disentangled representations:** Correlation matrix clustermaps for CIFAR-100, showing pronounced block diagonal structure, indicating a highly disentangled representation for MonoCon. By contrast, weak and diffuse correlations indicate entangled representations for baseline and standard MLP head models.

Table 3: Performance of models on CIFAR-100 for different levels of representation compression.

| Dimension | Model | 5-NN Acc (% ↑) | Recall@1 (% ↑) | Recall@5 (% ↑) | PCA recon. error ($\times 10^{-3}$ ↓) |
|---|---|---|---|---|---|
| 128 | Baseline | 77.68 | 77.08 | 81.75 | 6.34 |
| | MonoCon | 77.01 | 74.18 | 82.78 | **0.02** |
| | Std. MLP head | 78.34 | 75.56 | 83.15 | 1.69 |
| 64 | Baseline | 77.82 | 76.31 | 82.32 | 14.30 |
| | MonoCon | 77.01 | 74.18 | 82.78 | **0.05** |
| | Std. MLP head | 78.32 | 75.04 | 83.33 | 6.54 |
| 16 | Baseline | 77.15 | 47.62 | 83.37 | 28.81 |
| | MonoCon | 76.99 | **74.24** | 82.77 | **1.36** |
| | Std. MLP head | 77.85 | 58.60 | 83.77 | 26.84 |

tations to the first 128, 64, and 16 PCA components (Table 3) [1]. Most remarkably, under extreme compression to 16 dimensions, Recall@1 suffers a catastrophic decline for the baseline and standard MLP head models, whereas MonoCon's performance remains stable across all performance metrics even under this aggressive 8x compression. Specifically, MonoCon's Recall@1 performance is $\sim$ 27% better than the standard MLP head model and $\sim$ 55% better than the baseline, demonstrating clear superiority in high-precision retrieval. Ultimately, this superiority stems from MonoCon's ultra-low PCA reconstruction error, which is consistently more than an order of magnitude lower than that of other models due to MonoCon's low intrinsic dimensionality.

### 5.5 TRAINING DYNAMICS REVEAL MONOCON'S EMBEDDING DISTILLATION PROCESS

To gain mechanistic insights into how MonoCon learns high-quality, efficient, and disentangled representations, we quantified the evolution of MonoCon's representation over the entire course of a CIFAR-100 training run. Fig. 3 shows feature correlation matrix clustermaps for encoder output embeddings, as well as normalized and unnormalized embeddings generated by the monotonic MLP head. The clustermaps show strong correlations in the encoder's features in the initial stages, followed by a prolonged phase of systematic feature entanglement. The monotonic MLP head output has a remarkably simple block structure at the beginning of training, which becomes progressively more refined with restructuring of old feature blocks and inclusion of new ones. This qualitative evolution of clustermaps can be quantified using the effective dimensionality of embeddings, which

---

[1]The small differences in performance metrics across Table 2 and Table 3are due to the subtle effects of PCA-based truncation, which can result either in improvement due to denoising, or degradation due to information loss, depending on the model, metric, and level of truncation.

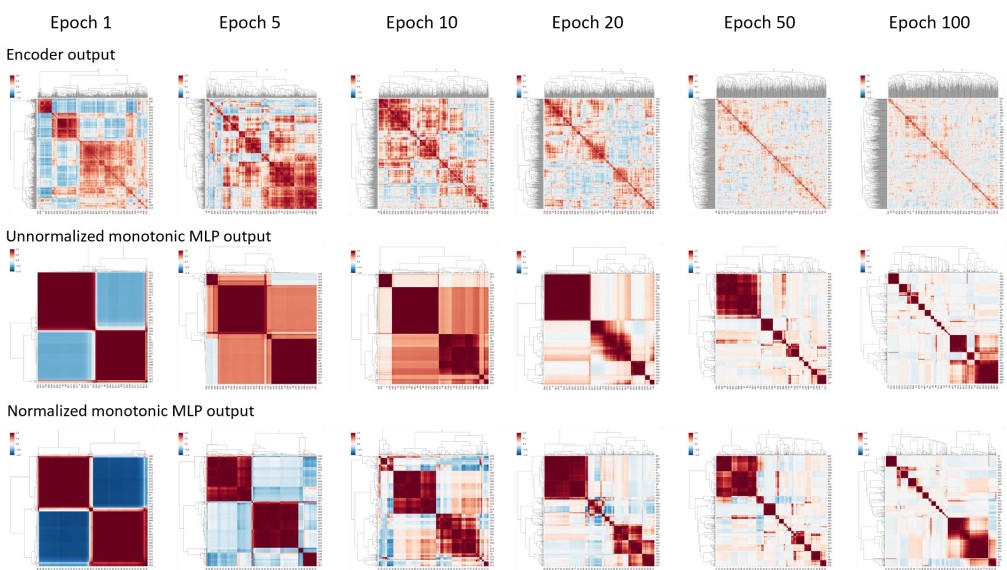

Figure 3: **Training dynamics reveal self-organized emergence of disentangled representations:** Correlation matrix clustermaps for encoder output (top row), unnormalized monotonic MLP head output (middle row), and normalized monotonic MLP head output (bottom row) at representative points during training. Feature correlation matrices were computed using embeddings from a fixed set of 1000 training images. Colorbar axis ranges from -1 to 1 for all plots.

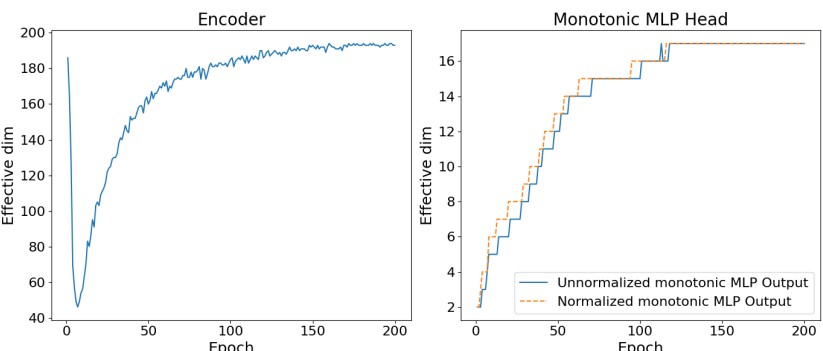

Figure 4: **Embedding distillation in action:** Effective dimensionality versus epoch number for the encoder output embeddings (left) and unnormalized (blue) as well as normalized (orange) monotonic MLP head output embeddings (right), for a CIFAR-100 training run.

can also be interpreted as the effective rank of feature correlation matrices. The effective dimensionality plots show a dramatic partial dimensional collapse of the encoder's representation in the initial phases of training, followed by a gradual recovery. Consistent with the clustermaps in Fig. 3, the monotonic MLP head's output rank increases systematically throughout training from 2 to 17 [2].

While one would naively have expected MonoCon's compactness to emerge through a top-down winnowing of the encoder's output representation, our findings reveal a highly counterintuitive bottom-up approach to representation learning. The training dynamics uncover a complex co-adaptation process driven by combined pressure exerted by the hard monotonicity constraint and

---

[2]The effective rank for this CIFAR-100 run (17) is higher than that reported for the run in Table 1 (14) due to the difference in validation frequency between the two runs, which influences early stopping. This higher $d_{\text{eff}}$ for the run shown in Fig. 4 also leads to higher 5-NN accuracy of 77.69%. This is consistent with MonoCon's bottom-up approach to building representation complexity.

the soft optimization objective. Once unfrozen after a head warmup phase prior to main training (Appendix A), the encoder initially reacts to the monotonicity "shock" by partially collapsing its representation, while the monotonic MLP head acts as a highly selective gatekeeper that prevents most features from passing through. As training progresses, the signal from gradients of the SupCon loss enrich the encoder's feature representation, while the Monotonic MLP head refines and combines these features into an increasingly structured and expressive representation. We call this self-organized learning process "embedding distillation", in analogy with the distinct and well-known technique of knowledge distillation Hinton et al. (2015).

## 6 DISCUSSION

By introducing MonoCon, we have shown that the inductive bias provided by a simple functional constraint, namely monotonicity, can generate ultra-compact high-fidelity representations in a completely self-organized manner. These representations provide robust and competitive performance, even under extreme representation compression. By analyzing feature correlation matrices, we showed that MonoCon's performance draws from a sophisticated form of disentanglement at the level of correlated groups of features.

The training dynamics elucidate the embedding distillation process, where the encoder's features are initially pruned by the functional constraint and later shaped by the optimization objective, with the monotonic MLP head serving as a feature refiner and blender, as well as an adaptive information bottleneck Tishby et al. (2000). While knowledge distillation creates compact *models* by using a student to distill the teacher's knowledge, embedding distillation creates compact *representations* by using an intelligent head to distill the information contained in the encoder's embeddings. The two are thus entirely separate, and potentially complementary techniques that could be used synergistically in future studies. A fascinating consequence of embedding distillation is that MonoCon's training curve itself represents a performance-efficiency tradeoff due to the monotonic increase in effective dimensionality (Fig. 4), which can be navigated at will using early stopping alone.

The chief limitation of MonoCon is that it often sacrifices a small amount of performance for gains in efficiency. In future work, it will be interesting to explore ways to soften the monotonicity constraint in a way that can potentially result in pareto improvement rather than performance-efficiency trade-offs. Other mathematical constraints such as convexity Amos et al. (2017) or equivariance Thomas et al. (2018) can also be incorporated as extensions of the MonoCon framework. The framework can also be extended beyond vision and NL tasks to other data modalities.

Since MonoCon relies on functional constraints rather than architectural or optimization ones, it is inherently complementary to all the approaches to representation and model compactness discussed earlier. Thus, there is tremendous potential for synergistically combining them with existing state-of-the-art tools for efficient AI. A particularly exciting frontier would be to train "MonoConized" versions of foundation models whose compact, disentangled representations could have significant implications for true on-device AI. Finally, while we have focused on efficiency, MonoCon's disentangled representations, and the hierarchical grouping of features into concepts Okawa et al. (2023); Gokhale (2023), also have massive implications for interpretability Wetzel et al. (2025) and explainable AI Dwivedi et al. (2023).

## 7 CONCLUSION

The need for efficient AI is paramount, whether from a computational, energetic, environmental, or financial perspective. MonoCon presents a simple and generalizable solution to a critical bottleneck in this challenging problem: representation compactness. Concretely, we have achieved nearly 9x reduction in effective dimensionality on a CIFAR-100 task and 3.4x reduction on an SNLI task, while retaining 99% of the baseline's performance. MonoCon achieved these results through a self-organized embedding distillation process, rather than top-down engineering. In a broader context, MonoCon provides a definitive proof of concept for a third paradigm, namely functional constraints, that can be blended with architectural and optimization constraints to forge a path towards learning highly efficient and intelligent representations.

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

## A    EXPERIMENT IMPLEMENTATION DETAILS

All models were trained on a machine equipped with an NVIDIA GeForce RTX 4050 GPU, an Intel Core i7-13700H CPU, and 16 GB of RAM. All models were trained using an AdamW optimizer Loshchilov & Hutter (2017), cosine annealing scheduler Loshchilov & Hutter (2016), weight decay Krogh & Hertz (1991), and gradient norm clipping Pascanu et al. (2013). For reproducibility, the random number generator seed was set to 42, and deterministic algorithms were used for cuDNN backend processes. For stable training, we used a weight decay parameter of $10^{-4}$ and gradient clipping norm of $1.0$. Before main training, we implemented a warmup phase with frozen encoder weights and monotonic MLP head learning rate of $10^{-4}$, for 10 epochs for vision tasks and 1 epoch for the NL task.

For CIFAR-100 runs, we used a mini-batch size of 256, SupCon loss temperature $\tau = 0.1$, and a learning rate of $2 \times 10^{-4}$ for the encoder as well as the head. The model was trained for a maximum of 200 epochs with early stopping triggered based on 5-NN accuracy after a patience of 20 epochs. For CIFAR-10, all hyperparameters were unchanged, except for the common learning rate for encoder and head, which was $5 \times 10^{-4}$. Validation metrics were computed every 5 epochs for all CIFAR experiments. For the NL task run, we used a mini-batch size of 128, $\tau = 0.05$, encoder learning rate $= 2 \times 10^{-7}$, and head learning rate $= 2 \times 10^{-4}$. The model was trained for a maximum of 40 epochs with early stopping triggered after a patience of 10 epochs based on the STSb score (Spearman coefficient), which was calculated after every epoch. For all models, the corresponding ablation studies were performed with unchanged hyperparameters.

## B    CHOOSING HYPERPARAMETERS

The core philosophy of MonoCon is to attach a small head to a powerful encoder. Thus, the primary contribution of the head is to provide an inductive bias through the monotonicity constraint rather than to add raw computational power. For performance-oriented experiments, we recommend using a more powerful encoder instead of increasing the size of the head. Increasing the depth of the monotonic MLP head from 1 to 2 hidden layers led to a severely over-compressed representation, and a massive performance drop in 5-NN accuracy for CIFAR-100. Overall, the performance is much less sensitive to hidden layer width. While we have not performed extensive hyperparameter sweeps, we found $d_{\text{hidden}} = 2d_{\text{enc}}$ to be a reasonable choice that is large enough to provide expressivity, yet small enough to avoid overfitting during training. Here, $d_{\text{enc}} = 512$ for ResNet34, and $d_{\text{enc}} = 384$ for the MiniLM.

The choice of differential learning rates for encoder and head is highly context dependent. For CIFAR-100, where the pre-trained ResNet34 performs poorly on CIFAR-100 classification (5-NN $\approx 18\%$), we found that full-fledged co-adaptation at the same learning rate yielded the best results. Indeed, fine-tuning the encoder at a 10x lower learning rate compared to the head led to a noticeable drop in 5-NN accuracy. On the other hand, the pre-trained off-the-shelf MiniLM has an STSb score of 82.03%, which outperforms the fine-tuned baseline. In this case, performance degrades significantly if the encoder is trained with the same (higher) learning rate as the head, and keeping the encoder learning rate as small as possible is the best choice.

## C    LLM USAGE STATEMENT

The author used Gemini 2.5 Pro during multiple aspects of the research. In the spirit of author contributions, the most concise and accurate summary of LLM usage in this paper can be expressed as follows: Author designed research. Author and Gemini performed experiments and analyzed data. Author wrote the paper with assistance from Gemini for editing and literature review. A more detailed breakdown is as follows:

- Project design: The author is the sole creator of the MonoCon framework, including the central idea of utilizing functional constraints, and specifically monotonicity, within a deep metric learning framework. The author is also solely responsible for strategic decisions such as choice of benchmarks, performance evaluation metrics, and analysis of training dynamics.

- Experiments and data analysis: The author made significant use of Gemini 2.5 Pro, particularly during the initial stages of the project, for rapid prototyping and iterative refinement. Gemini was used for code generation and as a sounding board for converging on key implementation details such as the choice of data augmentations and loss function. The main purpose of these initial efforts was to ensure strong baseline performance on CIFAR-100 image classification. Code generation was also used for computing performance metrics for data analysis. For all experiments and analyses, the final decision on implementation was made by the author.

  After the prototyping and refinement phase, the author personally reviewed, refactored, and commented Gemini generated code into a finalized set of jupyter notebooks. All data and performance metrics reported in this work were generated using these finalized scripts alone, which the author has provided as supplementary material.

- Manuscript writing: The author wrote the manuscript and sought Gemini's assistance in finding typos and polishing the text. The author also asked Gemini to suggest references for the section on related work, but independently ascertained the veracity and relevance of suggested references before citing them.

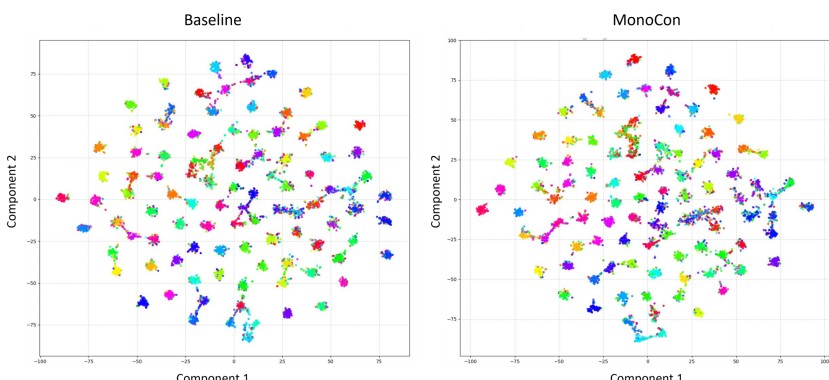

Figure 5: **Visualization of CIFAR-100 class separation:** The tSNE plots for baseline (left) and MonoCon (right) show a comparable level of class separation, consistent with comparable numbers for 5-NN accuracy.

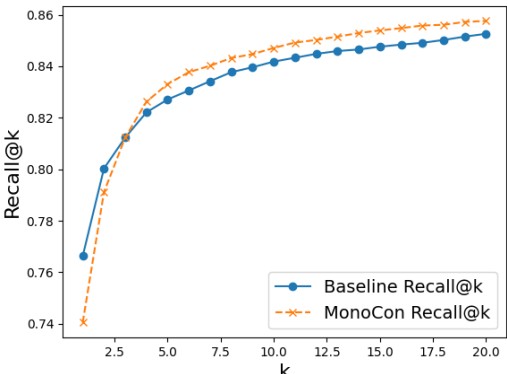

Figure 6: **Crossover in Recall@k vs k for CIFAR-100:** Baseline outperforms MonoCon on Recall@1 and Recall@2, but MonoCon shows superior performance on Recall@k for all $k > 3$, indicating superior global semantic structure.

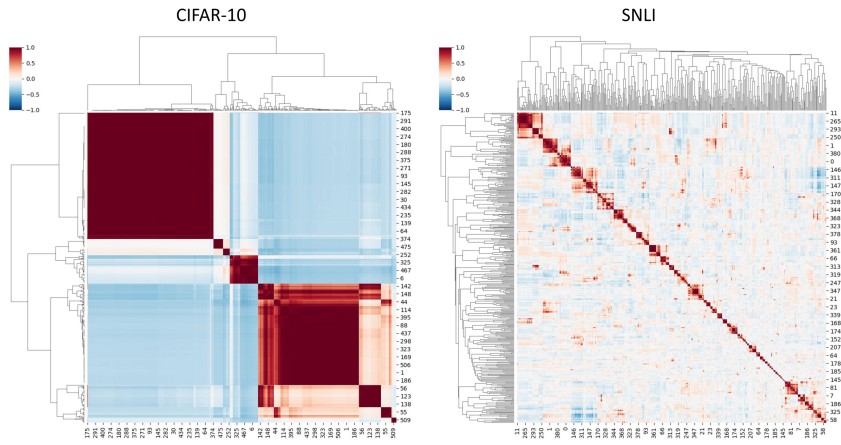

Figure 7: **Block diagonal structure in MonoCon's CIFAR-10 and SNLI representations:** Clustermaps and dendrograms for MonoCon's normalized output feature correlation matrices for CIFAR-10 image classification (left) and SNLI sentence similarity (right) tasks.