# OpenReview forum: "MonoCon: A general framework for learning ultra-compact high-fidelity representations using monotonicity constraints"
_ICLR.cc/2026/Conference — ICLR 2026 Conference Withdrawn Submission_

### Official Review · Reviewer_scD9 · 2025-11-01

**Soundness:** 3
**Presentation:** 3
**Contribution:** 2
**Rating:** 2
**Confidence:** 4

**Summary:**

- MonoCon attaches a monotonic MLP head to a pretrained encoder and trains end-to-end with SupCon; the head’s output dimensionality matches the encoder and is kept at test time.
- Monotonicity is enforced via non-negative weights, aiming to make representations compact and stable.
- Benchmarks: CIFAR-10/100 and SNLI→STSb; metrics: 5-NN/Recall@k plus PCA-based proxies (effective dimensionality, reconstruction error).
- Results: achieves large dimension reduction and lower reconstruction error with minimal accuracy loss, supported by correlation-structure analyses.
- Positioning: uses monotonic constraints not for compliance or calibration, but as a general head for representation compression/regularization.

**Strengths:**

- Simple and general: a lightweight, plug-on head for existing encoders with a clear training/inference pathway.
- Consistent compression: substantially reduces effective dimensionality while retaining accuracy across both vision and language tasks.

**Weaknesses:**

- Missing comparisons along two crucial axes. To attribute gains to monotonicity (rather than capacity/regularization), the paper should compare monotonic MLP (proposed) vs DLN (lattice-based monotonic) to isolate the unique effect of monotonicity and to quantify the practical trade-offs (accuracy–compression–latency) under identical training and capacity.  In parallel, it needs (B) non-monotonic compact-representation baselines, (e.g., VICReg, Barlow Twins) to show added value over alternative compression strategies. Without both axes, it remains unclear whether improvements stem from monotonicity itself or from architectural/regularization choices.
- Limited backbone/scale generalization: vision focused on ResNet, NLP on MiniLM, and small datasets (CIFAR, SNLI→STSb); needs tests on ViT/Swin/ConvNeXt, BERT/RoBERTa, and larger benchmarks (ImageNet/COCO, GLUE).
- Proxy-metric dependence: heavy reliance on PCA-based proxies; lacks task-level robustness (e.g., OOD kNN, noise/corruption settings).
- Statistical reliability: no multi-seed mean±std, making stability/variance unclear.

**Questions:**

- Can you compare monotonic MLP (proposed) vs DLN (lattice-based monotonic) to isolate the unique effect of monotonicity and to quantify the practical trade-offs ?
- Do the performance–compression trade-offs hold across more backbones and larger datasets (ViT/Swin/ConvNeXt; BERT/RoBERTa; ImageNet/COCO, GLUE)?
- Will you report ≥3-seed mean±std for key metrics to establish stability?
- Beyond PCA proxies, can you evaluate task-level robustness (OOD kNN, noise/corruption, downstream retrieval/classification) to verify the benefit of the monotonic head?

---

### Official Review · Reviewer_3mnW · 2025-11-01

**Soundness:** 1
**Presentation:** 1
**Contribution:** 1
**Rating:** 0
**Confidence:** 4

**Summary:**

This paper introduces MonoCon, a framework that extends deep metric learning encoder with a trainable MLP head. To enforce monotonicity, the weights of this head are squared, and it utilizes Leaky ReLU activation function. The authors claim these constraints produce more compact and robust representations compared to either the direct encoder outputs or an unconstrained MLP head. The method is evaluated on two vision datasets (CIFAR-10, CIFAR-100) and one NLP dataset (SNLI).

**Strengths:**

I don't see any notable strengths in the current form of the submission.

**Weaknesses:**

**W1)** A major weakness is the absence of a theoretical grounding or clear intuition. The paper asserts that monotonicity leads to more compressed and robust embeddings but provides no formal analysis or convincing explanation for  why  this constraint should produce such benefits.

**W2)** The authors evaluate on the CIFAR training set (line 208) rather than the standard test split, which fails to measure generalization. Constraining the network with monotonicity potentially helps against overfitting that the baseline experiences. The authors should evaluate in a proper setting that verifies generalization of the method. Larger-scale metric learning benchmarks, and ideally a second task, such as image retrieval should be added to truly confirm the useful properties of MonoCon.

**W3)** The empirical results are unconvincing. In the low-dimensional regime (Table 3, dim=16), MonoCon underperforms the baseline on two out of three metrics. Furthermore, the deff​ metric appears disconnected from task performance: the baseline performs better at dim=16 despite a much higher $d_{eff}​$ (125), which questions the metric's practical relevance. To demonstrate broader applicability, the evaluation should also include modern architecture, such as vision transformer.

**W4)** The paper fails to adequately position its contributions and lacks essential baseline comparisons.
- **Novelty:** The discussion of novelty is insufficient. The paper does not clearly differentiate its contributions from prior works that apply monotonicity constraints to deep networks, such as \[1\].
- **Baselines:** The paper mentions standard compression methods (e.g., PCA, product quantization) in the related work but provides no empirical comparison. Without these baselines, it is impossible to assess the method's effectiveness relative to established techniques.

\[1\] Deep lattice networks and partial monotonic functions, You et al., NeurIPS, 2017

**W5)** Although the extent of the use of LLM is disclosed in the Appendix, its heavy use makes reading the paper text more difficult. Factual information is often buried in convoluted sentences and vague language. For example, the sentence, "We implement a differential learning rate strategy to flexibly switch between fine-tuning the encoder at a lower rate, to full-fledged co-adaptation between encoder and head at the same rate," is ambiguous and lacks concrete details about the switching mechanism or the rates used. This convoluted style hinders readability and comprehension.
Additionally, the in-text citations lack brackets, causing them to blend with the text and deviate from the author guidelines.

**Questions:**

N/A

---

### Official Review · Reviewer_EV8x · 2025-11-03

**Soundness:** 2
**Presentation:** 2
**Contribution:** 2
**Rating:** 2
**Confidence:** 4

**Summary:**

MonoCon offers a simple yet generalizable solution to a critical bottleneck in representation compactness. Specifically, the authors demonstrate that MonoCon achieves up to a 9× reduction in effective dimensionality on the CIFAR-100 task and a 3.4× reduction on the SNLI task, while preserving 99% of the baseline performance. These impressive results are obtained by incorporating a lightweight monotonic multi-layer perceptron (MLP) head that can be seamlessly attached to any pre-trained encoder.

**Strengths:**

1. Effectiveness and generality of MonoCon: The proposed MonoCon framework is both simple and remarkably effective on the CIFAR-100 and SNLI tasks. The methodology achieves a substantial reduction in effective dimensionality—by up to 3.4× on SNLI—while maintaining 99% of the baseline performance. This demonstrates that the approach successfully balances representation compactness and accuracy. More importantly, the design appears architecture-agnostic, suggesting that the monotonic constraint can be seamlessly integrated into various model architectures, potentially broadening its applicability across domains.

2. Insights from training dynamics: The analysis of training dynamics uncovers a self-organized co-adaptation between the encoder and the classification head, characterized by a rapid early-stage dimensional collapse followed by a gradual recovery phase. This finding is particularly intriguing, as it reveals the internal coordination between model components under monotonic constraints. The authors’ effort to visualize and interpret this phenomenon provides valuable insight into the representational evolution during training.

**Weaknesses:**

1. **Motivation — Why does monotonicity work intuitively?** This is the key question the authors should have addressed, or at least attempted to answer, in this work. However, the paper lacks a dedicated section exploring the underlying intuition or theoretical rationale for why monotonic constraints improve learning. While the section on training dynamics briefly touches on this aspect, it merely reports empirical phenomena without delving deeper into *why* such behavior emerges. A more thorough analysis or discussion linking monotonic learning to potential cognitive constraints in human learning or mechanisms in the human brain would significantly strengthen the paper’s contribution.

2. **Limited empirical validation.** Although the proposed idea is simple and appears effective on CIFAR-100 and SNLI, the empirical analysis is limited in scope. The authors should evaluate the proposed monotonic constraint across multiple architectures to verify whether the observed trend is consistent and reproducible under different model designs. Such validation would greatly enhance the generality and credibility of the findings.

3. **Insufficient task diversity.** The experimental evaluation is currently restricted to CIFAR-100 and SNLI. To better demonstrate the effectiveness of the proposed representation compactness, additional experiments on larger and more diverse classification datasets, such as ImageNet, would be valuable. Broader evaluations across domains would help establish whether the approach truly scales and benefits memory efficiency in more complex, real-world settings.

4. **Impact on generalization.** While the results show only a small performance drop on the tested datasets, it remains unclear how representation compactness affects generalization, particularly in *out-of-distribution (OOD)* scenarios. A deeper analysis of whether compact representations improve robustness or transferability under distribution shifts would make the work more comprehensive and impactful.

**Questions:**

1. What is the underlying intuition behind the effectiveness of monotonicity in representation learning? Can the authors provide a theoretical or conceptual explanation for why imposing monotonic constraints leads to improved learning dynamics or compact representations?

2. Does the proposed monotonic constraint generalize across architectures? Have the authors tested whether the observed performance and dimensionality reduction trends remain consistent under different model architectures and design choices?

3. Can the proposed approach scale to larger and more diverse tasks? How does MonoCon perform on more complex datasets such as ImageNet or across domains beyond CIFAR-100 and SNLI? Does it maintain efficiency and performance advantages at scale?

4. How does representation compactness influence model generalization and robustness? Does the compact representation learned via MonoCon improve or degrade performance in out-of-distribution or transfer learning settings?

---

### Official Review · Reviewer_3PVx · 2025-11-09

**Soundness:** 2
**Presentation:** 3
**Contribution:** 1
**Rating:** 2
**Confidence:** 4

**Summary:**

This paper introduces MonoCon - a monoticity-constrained (squared weights fed into a ReLU) MLP attached to representation models then trained using contrastive objectives such as SupCon to produce more compact representation spaces; with results verified on CIFAR-10, CIFAR-100 and SNLI.

**Strengths:**

* The paper is well written and easy to follow; while the setup for MonoCon is easy to utilize.
* The recall capabilities for highly restricted representation spaces as shown in Tab. 3, at least on CIFAR100, is very convincing.
* I really like the study on compression and training dynamics conducted in order to understand practical impact of MonoCon; within the realm of extrapolatable context given the utilized datasets, the results are quite convincing.

**Weaknesses:**

I do not believe that this paper is ready (yet) for publication, for the following primary reasons

While one could argue that the contribution is too simplistic (squared weights + ReLU on a MLP head), I strongly believe that this could be a feature IF it would be generally applicable. Unfortunately, experiments were only conducted on very small-scale benchmarks and architectures (limited also only to convolutional networks for vision), and NO conclusion can be drawn how these insights hold up to any relevant scales useful for general representation learning scenarios (i.e. at the very least, studies on ImageNet would be great to see).

Moreover, the relevance and connection of using a monotonic reprojection is introduced pretty much ad-hoc, with at most some pointers to some related literature. It's prudent for the authors to provide significantly more context as to why monotonic reprojections can helo compress representations down to the degree seen.

**Questions:**

See Weaknesses

---

### Author Response · Authors · 2025-11-13

We thank all reviewers for their thoughtful reviews. In the light of these comments, we have decided to withdraw and improve the manuscript.

---

### Note · Authors · 2025-11-13

I have read and agree with the venue's withdrawal policy on behalf of myself and my co-authors.